# The Multi-Faceted Role of Gut Microbiota in Alopecia Areata

**DOI:** 10.3390/biomedicines13061379

**Published:** 2025-06-04

**Authors:** Andrea Severino, Serena Porcari, Debora Rondinella, Enrico Capuano, Tommaso Rozera, Francesco Kaitsas, Antonio Gasbarrini, Giovanni Cammarota, Gianluca Ianiro

**Affiliations:** 1Department of Translational Medicine and Surgery, Università Cattolica del Sacro Cuore, 00168 Rome, Italy; 2Department of Medical and Surgical Sciences, UOC CEMAD Centro Malattie dell’Apparato Digerente, Medicina Interna e Gastroenterologia, Fondazione Policlinico Universitario Agostino Gemelli IRCCS, 00168 Rome, Italy

**Keywords:** alopecia areata, microbiota, fecal microbiota transplantation

## Abstract

Alopecia areata (AA) is a complex autoimmune disorder with multifactorial pathogenesis. Recent research highlights the gut microbiota as a possible key player in AA pathogenesis through the gut–skin axis: gut dysbiosis may disrupt intestinal barrier integrity and immune tolerance by affecting T regulatory cells, potentially contributing to disease onset and progression. The purpose of this review is to analyze the current evidence on the correlation between gut microbiota and AA, dissecting both the pathogenetic role of its alterations in the onset and progression of disease and its potential role as a therapeutic target.

## 1. Introduction

Alopecia Areata (AA) is an autoimmune condition causing temporary, non-scarring hair loss while preserving the hair follicles [1]. Classified as a non-scarring primary hair loss disorder, AA shares this category with conditions like androgenic alopecia and telogen effluvium [2]. Several subtypes of AA have been described, according to clinical pattern and hair loss extension. Subtypes vary from small well-defined patches of hair loss (patchy AA), considered the most common form [3], to the diffuse involvement of the scalp (alopecia totalis) or the entire body (alopecia universalis) [4,5].

A recent systematic review and meta-analysis of 94 studies estimated the global prevalence of AA to be around 2%. The data revealed a considerable increase in prevalence over time: 1.02% before 2000, 1.76% between 2000 and 2009, and 3.22% after 2009. Prevalence rates also varied widely across different geographic regions [6]. Additionally, AA was found to be more prevalent in the pediatric population (1.92%) compared to adults (1.47%), but no significant sex-related differences have been identified in the prevalence of alopecia areata [6].

Alopecia areata’s development is unpredictable; about 80% of those newly affected may experience hair regrowth within a year. However, the condition usually follows an inconsistent pattern of relapses and remissions [7]. Several factors could be recognized as associated with a poor outcome of AA, such as extent and duration of hair loss, age at onset, and family history [8]. The extensive hair loss in AA is directly associated with a poor prognosis [8], with a progression rate of nearly 5% from limited hair loss to complete scalp loss (alopecia totalis) or total body hair loss (alopecia universalis) [8]. Moreover, clinical variants like ophiasis, sisaipho, and Marie Antoinette syndrome are linked with worse outcomes [2,9]. Nail involvement—observed in 15–44% of cases—has also been associated with more severe disease forms [10,11].

AA is often accompanied by several comorbidities, including atopic dermatitis, thyroid disorders, lupus, celiac disease, metabolic syndrome, and various psychiatric conditions [12]. These associations highlight the complex immunological background of the disease, further supporting the extensive multifactorial immune dysregulation in the pathogenic mechanism of AA. However, the exact pathogenesis of alopecia areata remains incompletely understood. It is believed to involve a collapse of immune privilege in hair follicles, leading to an autoimmune T-cell-mediated attack. Both genetic predisposition and environmental triggers appear to contribute to this aberrant immune response [13,14].

This pathogenic complexity is also reflected in the therapeutic landscape. The therapeutic management of AA remains challenging and highly variable, depending on the severity and extent of the disease. While mild cases may respond to topical or intralesional corticosteroids, more severe or resistant cases often require systemic agents such as JAK inhibitors, immunosuppressants, or biologic therapies [3,15].

In recent years, emerging evidence has increasingly supported a possible link between gut microbiota and the onset, progression, and treatment response of autoimmune diseases, including alopecia areata. The gut–skin axis—defined as the bidirectional communication between the gut microbiome and the skin—has gained considerable attention as a potential player in the immunopathogenesis of AA [16]. Alterations in the composition or function of gut microbiota may influence systemic immune responses, potentially exacerbating autoimmune processes in genetically susceptible individuals [17,18]. Understanding the multi-faceted interplay between gut microbes and host immunity could unlock new insights into AA’s pathogenesis and pave the way for innovative therapeutic strategies [19,20].

## 2. Immune Privileges and Their Collapse in Alopecia Areata

Immune privilege (IP) describes specialized anatomical sites in the body where immune activity is tightly regulated to prevent damage to vital or regenerative tissues [21]. The hair follicle, particularly during the anagen (growth) phase, is a well-characterized immune-privileged structure, supported by multiple mechanisms that actively suppress immune surveillance and inflammation. These include low expression of MHC class I molecules, the absence of MHC class II, the secretion of local immunosuppressive cytokines such as TGF-β1, α-MSH, and IGF-1, and the expression of checkpoint molecules like PD-L1, which inhibit T-cell activation [22]. Collectively, these features establish a localized immune-tolerant environment that shields the follicle from immune-mediated injury, ensuring its normal function and structural integrity.

In alopecia areata, this immune privilege collapses. Studies have shown an aberrant upregulation of MHC class I and II molecules on hair follicle epithelium, enabling the presentation of autoantigens to immune cells [13]. This leads to infiltration by autoreactive CD8+NKG2D+ cytotoxic T lymphocytes, which are thought to play a central role in follicular destruction, as well as natural killer (NK) cells and antigen-presenting cells like dendritic cells [14,22]. Pro-inflammatory cytokines, notably interferon-gamma (IFN-γ) and IL-15, further drive the inflammatory cascade, amplifying the autoimmune response through the JAK-STAT pathway [19,23]. As a result, the anagen hair follicle prematurely transitions to catagen or telogen phases, clinically manifesting as sudden hair loss.

Interestingly, the collapse of immune privilege is not a one-time event but may occur cyclically, aligning with the relapsing-remitting nature of the disease. The possibility of restoring or reinforcing immune privilege pharmacologically—through PD-L1 agonists, JAK inhibitors, or cytokine modulation—represents a promising therapeutic frontier in the management of AA [19,23].

## 3. Gut Microbiota: Composition and Functions

The gut microbiota is nowadays considered the largest microbial community in humans, gathering more than 100 trillion microorganisms [24,25]. Compared to the human genome, which only contains roughly 23,000 genes, the gut microbiome encodes over 3 million genes and generates hundreds of metabolites [26,27]. The composition of each person’s gut microbiota is formed early in life as a result of newborn transitions (birth gestational date, delivery procedure, milk feeding technique, weaning phase) and environmental influences, including antibiotic use [26,28]. This unique and healthy core native microbiota is maintained throughout adulthood and varies from person to person depending on enterotypes, BMI level, frequency of exercise, lifestyle, and cultural and nutritional preferences [26]. This stability is thought to affect approximately 85% of the gut microbiota [29], while the remaining 15% varies in response to the aging process [30] and environmental changes, and it represents part of our ability to adapt to those changes themselves. It has also been shown that intestinal microbiota biogeography and metabolome undergo circadian oscillations [29,31].

All the three kingdoms of life, Bacteria, Archaea, and Eukaryotes, are included in the composition of the gut microbiota [32]. Among Bacteria, the most represented phyla are Firmicutes and Bacteroidetes followed by Actinobacteria, Proteobacteria, and Tenericutes [29,33]. Among Eukarya, some *Candida* spp. are the most prevalent, whereas the Euryarchaeota is the most represented phylum belonging to Archaea [33]. In particular, 95% of the taxonomy at the phylum level can be traced back to the summation of Firmicutes and Bacteroidetes [29]. Individually these two different phyla show a complementary trend and the growth of one corresponds to the decrease in the other. Yet, the functionality expressed in terms of main metabolic pathways does not seem to reflect this difference, demonstrating that, at least in the main metabolic pathways, bacterial genomes appear to be mostly shared between these two phyla [29,34].

Moreover, the gut microbiota is geometrically organized along the longitudinal and cross-sectional axes of the gut [32,35]. Longitudinally, the gut environment varies considerably between distinct anatomical regions in terms of physiology, pH, oxygen tension, host secretions, and substrate abundance [36]. For these reasons, bacteria pass from approximately 100 per gram near the duodenal papilla to approximately 100 billion per gram in the distal colon [29,32]. From a taxonomic point of view, in the small intestine, Firmicutes and Proteobacteria are dominant, while in the colon the main phyla are Firmicutes and Bacteroidetes [29,32]. Radially, as we move from the lumen towards the periphery of the section, a decrease in *Bacteroidaceae* in favor of *Ruminococcaceae* and *Lachnospiraceae* can be observed. It has also been demonstrated that two separate ecosystems with diverse metabolic and immune activities are present in the luminal microbiota and in the mucosal-associated microbiota [37], with a prevalence of *Akkermansia* and *Bacteroides* genres in the last one [29].

From a functional point of view, more and more studies show that the human microbiota is closely involved in nutrient extraction, metabolism [25,38], and immunity [39]. The gut microbiota plays a critical immunological role in preventing the colonization of pathogenic bacteria by limiting their ability to grow, consuming available resources, and/or producing bacteriocins. By preserving the integrity of the intestinal epithelium [35,40], gut microbiota also prevents bacterial invasion [26,41]. The competition mechanisms used by microorganisms to avoid harmful colonization include food metabolism, pH alteration, the production of antimicrobial peptides, and impacts on cell signaling pathways [26]. Furthermore, recent research has revealed that commensal bacteria and their metabolites play a crucial part in controlling the growth, homeostasis, and function of innate and adaptive immune cells [26,39,42].

## 4. Gut-Skin Axis and Its Role in the Pathogenesis of Alopecia Areata

### 4.1. Gut Microbiota and Immune System

In optimal conditions, the gut microbiome performs numerous essential functions for the well-being of the holobiont [43]. These functions range from extracting energy from food through enzymatic activity to supplying the host with crucial vitamins, such as vitamin K and several B complex vitamins. Moreover, the intestinal flora provides protection against external pathogens, aids in maintaining the integrity of the intestinal epithelial barrier, and is crucial for the functional development of the gut immune system [44]. Additionally, the gut microbiota generates various metabolites that act as local and systemic signaling molecules, significantly impacting health and disease states [45]. Among the most significant products of the gut microbiota are short chain fatty acids (SCFAs), compounds which play a critical role both locally and systemically. Locally, SCFAs serve as the main energy source for colonic epithelial cells and regulate energy metabolism while also exerting immunomodulatory effects to maintain the balance between anti-inflammatory and pro-inflammatory responses [44]. Biliary acids (BAs) are another crucial component in the interaction between gut microbiota and host. Research has demonstrated their immunomodulatory effects, promoting an anti-inflammatory M2 macrophage phenotype while reducing the pro-inflammatory M1 phenotype in both the gut and liver. BAs also decrease IFNγ and IL-6 and increase IL-10 levels [46]. Studies have further revealed the role of secondary BAs in modulating the adaptive immune system, primarily through regulating the Treg/Th17 ratio and Th1 populations [43]. The intestinal barrier is one of the most vital internal barriers in the human body. Its primary role is to facilitate the absorption of essential nutrients and fluids while preventing the entry of harmful substances. Persistent disruption of the gut barrier can allow microbial components to enter the body, resulting in systemic low-grade inflammation [45].

### 4.2. Gut–Skin Axis

In the last decade, substantial evidence has demonstrated that the gut–skin relationship is intimate and bidirectional [47,48]. To date, the precise mechanisms by which the gut microbiota affects skin health remain incompletely understood [49]; however, evidence suggests that metabolic and immunological effects of gut microbiota can influence skin diseases [48]. The integrity of the intestinal barrier, in conjunction with the action of mucus, immune cells, IgA, and antimicrobial peptides (AMPs) produced by epithelial cells, prevents the entry of gut bacteria into the bloodstream [50]. Limiting microorganism contact with the gut epithelial membrane to reduce inflammatory responses and microbial translocation is crucial for maintaining the host’s homeostatic balance [49]. Specific metabolic products of gut microbes can directly influence normal physiology and disease processes: microbial communities maintain gut barrier integrity primarily by converting indigestible complex polysaccharides into vitamins and SCFAs [51,52]. Disruption of gut integrity, imbalance within microbial communities, and modification in bacterial metabolic products can have a significant impact on the overall homeostasis of skin [53]. Skin regeneration and keratinization are both regulated by specific transcriptional processes: the gut microbiome influences skin homeostasis by affecting the signaling processes that maintain epidermal differentiation [54]. Dysbiosis in the gut is known to contribute to three common skin disorders: psoriasis, atopic dermatitis, and acne [53]. There are also reports on the association of gut dysbiosis with some less common diseases, such as alopecia areata [49].

### 4.3. How Does the Gut Microbiota Influence the Pathogenesis of Alopecia Areata?

An important mechanism in the pathogenesis of alopecia areata is the modulation of functionality of T-lymphocytes Treg. Numerous studies have supported the influence of SFCAs on the number and functionality of intestinal Treg [55]. The SCFAs bind to different G protein-coupled receptors (GPCRs) on immune and gut epithelial cells: the best characterized GPCRs that respond to SCFAs are GPR43, GPR109, and GPR41. Through interaction with these receptors located on the surface of Tregs, SCFAs facilitate the proliferation of Tregs already present in the colon, augment the expression of Foxp3 and interleukin-10 (IL-10) within Tregs, and enhance their suppressive function [55,56]. Exacerbated or unresolved inflammation has been observed in GPR43-deficient (Gpr43-/) mouse models of colitis, arthritis and asthma [57], indicating the influence of this receptor in aggravating inflammatory disease. In addition, SCFAs inhibit histone deacetylases (HDACs), promoting histone acetylation and Treg gene activation [55]. A gut dysbiosis with a depletion of short-chain fatty acid (SFCA)-producing bacteria could result in a reduction in the number and functionality of T-lymphocytes Treg, promoting development of the disease in genetically susceptible subjects. In this context, the gut microbiota could be implicated in the pathogenesis of alopecia areata [58]. Another hypothesis regarding the relationship between gut microbiota and alopecia areata posits that a pathological state, known as “leaky gut” syndrome (LGS), could serve as a trigger for immune homeostasis disruption. LGS is characterized by increased intestinal permeability caused by dysfunction of the intestinal barrier, with disruption at the level of the tight junctions (TJs) or damage to the mucosal layer [59]. In this state, translocation of commensal and pathogenic bacteria occurs, inducing systemic inflammation. However, to date, evidence of association with increased intestinal permeability in these patients is limited and equivocal. A recent study (2022) investigated intestinal permeability in 70 alopecia areata patients and compared them to 70 healthy controls: no significant differences in serum zonulin (a TJ protein) levels were found [60]. In contrast, a case–control study published in 2023, which included 50 patients with alopecia areata and 30 healthy controls, identified a correlation between elevated plasma concentrations of claudin-3—an essential component of tight junctions that may serve as an indicator of intestinal barrier integrity—and the severity of AA disease [61]. Figure 1 summarizes the main gut-microbiome mediated mechanisms involved in the pathogenesis of AA.

## 5. Gut Microbiota in Alopecia Areata

### 5.1. Evidence of Correlation Between Gut Microbiota and Alopecia Areata

In light of these pathophysiological observations and clinical hypotheses emerged exploring the connection between gut microbiota and other systemic autoimmune diseases and immune-mediated skin diseases, several studies on the microbiota of AA patients have been conducted in recent years [58]. Research conducted on mouse models has provided substantial evidence suggesting a potential link between gut microbiota and alopecia areata. In 2003, McElwee et al. [62] transplanted skin from mice naturally affected by alopecia areata onto healthy mice models. These grafted mice were given a diet rich in soy oil, and a large proportion did not develop alopecia areata, indicating that diet might play a role in controlling skin inflammation. In 2017, Nair et al. [63] transplanted skin from alopecia areata-affected mice onto unaffected mice, which led to the disease manifesting in the previously healthy mice. Prior to this, the engrafted mice had been treated with a broad-spectrum antibiotic combination. The mice that received antibiotics were shielded from developing alopecia areata, whereas those that did not receive treatment experienced hair loss. Furthermore, a reduction in skin-infiltrating T-lymphocytes CD8+NKG2D+ was noted in the group treated with antibiotics, suggesting that gut microbiota may influence the infiltration of T-cells into hair follicles, which disrupts their IP in AA. To date, evidence of the relationship between gut microbiota and AA in clinical studies is limited. The most significant data available regarding the potential role of gut microbiota modulation in AA are the observations of Rebello in 2017 [64] and Xie in 2019 [65], where hair repopulation was observed in alopecia areata patients after undergoing fecal material transplantation (FMT) for other indications such as *Clostridiodes difficile* infection or Crohn’s disease.

### 5.2. Analysis of the Gut Microbiota in Alopecia Areata

Based on these findings, several studies have characterized the gut microbiota of patients with alopecia areata (Table 1). Moreno-Arrones et al. [66], in 2020, conducted a study to determine if patients affected by alopecia universalis present differences in gut bacteria composition compared to healthy controls. Fifteen individuals with alopecia universalis were included in a study and compared to a group of 15 healthy individuals: while there were no statistically significant differences in the alpha or beta diversity between the two groups, patients with alopecia showed several qualitative changes in gut microbiota composition (Table 1). Additionally, the researchers sought to identify potential bacterial biomarkers for the disease, discovering that a 25% rise in the abundance of *Clostridiales vadin BB60* and *P. distasonis* was associated with a 9.4% and 11.4% increased risk of developing alopecia universalis, respectively. Another study conducted in 2021 [67] compared the gut microbiota of 33 AA patients with 35 healthy controls. Although no significant differences in α-diversity were found between the two groups, the microbiota structure differed significantly (Table 1). *Achromobacter*, *Megasphaera*, and *Lachnospiraceae incertae sedis* were identified as biomarkers to distinguish between patients with alopecia areata and healthy controls. In 2021, Rangu et al. [68] conducted a cross-sectional study extending the analysis to the pediatric population: 41 children aged 4–17 years with alopecia areata and 41 of their siblings without alopecia areata were included in the analysis. Compared to their sibling controls, the relative abundance of *Ruminococcus bicirculans* in patients with alopecia areata was decreased. More recently, Lee et al. conducted a cross-sectional study [69] comparing fecal samples collected from 19 AA patients and 20 healthy controls. The study did not find a statistically significant difference in terms of alpha-diversity; however, the qualitative composition of the gut microbiome differed between the two groups (Table 1). In 2024, researchers in Italy published findings from an observational cohort study that compared healthy adults with individuals diagnosed with AA [70]. The investigation revealed that AA patients demonstrated decreased richness and evenness. Through differential abundance analysis, the study identified specific microbial markers associated with AA, particularly *Firmicutes*, *Lachnospirales*, and *Blautia*. In contrast, *Coprococcus* was found to be more prevalent in healthy subjects. Table 1 summarizes the main features of the outlined studies (data regarding disease duration were reported, when available; data on the use of concomitant therapies were not available).

The observed differences in potential biomarkers across studies underscore the significance of geographical location as a confounding variable. This could be attributed to the distinct dietary habits prevalent in the regions where the studies were conducted, which may partially account for the variability observed. Additional factors that could contribute to bias and explain the substantial variability in the data across different studies include the employment of diverse sequencing methods, the relatively small sample sizes, and specific patient characteristics such as age, comorbidities, and concomitant medication use.

Based on this evidence, to investigate the potential causal relationship between gut microbiota and AA, several Mendelian randomization analyses have been conducted in recent years, suggesting a probable causality between gut microbiota and AA. Two distinct studies published in 2024 [71,72] identified specific taxa demonstrated to have a protective role and others with a causal role toward AA. However, the studies yielded non-univocal results, necessitating validation through further research.

Taxa identified as statistically significantly correlated to AA are presented in Table 2.

## 6. Gut Microbiota as Therapeutic Target in Alopecia Areata

Current evidence indicates that the gut microbiota significantly influences skin health and the development of skin diseases through the mechanisms previously described. In 2022, Mahmud et al. conducted a systematic review to examine the impact of various factors, such as diet, prebiotics, probiotics, and antibiotics, on the composition of the gut microbiota and how these changes may affect skin health and the onset of skin-related diseases [49]. In this context, it is plausible to hypothesize that the gut microbiota could serve as a therapeutic target in the treatment of AA. Due to the limited comprehension of the exact pathological mechanisms behind AA, existing treatments such as corticosteroids, other immunomodulators, and minoxidil show limited effectiveness and come with notable side effects and high rates of relapse. Consequently, investigating the connection between AA and gut microbiota could lead to new strategies for treatment and prevention, improving the management of AA [71]. However, the evidence in this area remains relatively limited, as the causal relationship between gut microbiota and alopecia areata is still under debate, and research efforts have predominantly focused on more established pathogenic pathways. Several studies in the literature have been conducted using animal models to investigate the role of probiotic and postbiotic supplementation as a therapy for AA [16]. Specifically, two studies by Kimoto-Nira in 2007 and Levkovich et al. in 2013 [47,73] demonstrated that probiotic supplementation with *Lactococcus lactis* subsp. *cremoris H61* and *Lactobacillus reuteri*, respectively, could benefit disease control in mouse models. Subsequently, in 2018, Borde and Astrand conducted multiple studies assessing the influence of propionate as a treatment for AA in mice, yielding conflicting results [56]. While studies analyzing the modulation of gut microbiota and its influence on the natural history of alopecia areata in animal models are not particularly abundant, the paucity is even more pronounced in human studies. The primary evidence available includes the aforementioned studies by Xie [65] and Rebello [64], which are two case reports where hair repopulation was observed in alopecia areata patients following FMT for other indications, such as *Clostridioides difficile* infection or Crohn’s disease.

## 7. Conclusions

AA is a complex autoimmune disorder characterized by a multifactorial pathogenesis that includes genetic predisposition, immune dysregulation, and environmental triggers. Recent evidence suggests that the gut microbiota may significantly contribute to the onset and progression of the disease, likely through its impact on immune homeostasis and systemic inflammation via the gut–skin axis. Although preclinical studies have indicated a role for gut dysbiosis in modulating immune responses that affect hair follicle immune privilege, clinical data remain limited and often inconsistent due to geographical, methodological, and dietary variations among studies. Nonetheless, specific bacterial taxa have been associated with either protective or causative roles in AA, and preliminary reports of therapeutic benefits from interventions such as FMT provide intriguing support for this hypothesis. Despite the growing interest and accumulating indirect evidence, the causal relationship between gut microbiota and AA has yet to be firmly established. The results of Mendelian randomization analyses are promising but require further validation through large-scale, longitudinal cohort studies, in order to more consistently identify a potential microbial signature characteristic of this condition and to precisely elucidate additional pathogenetic mechanisms involved. Similarly, interventional prospective studies and randomized clinical trials exploring microbiota-targeted therapies (such as probiotics, prebiotics, postbiotics, and dietary interventions) are necessary to assess their safety and efficacy in reshaping gut microbial composition, and long-term impact on the disease course. In consideration of the highlighted findings, it is evident that we remain distant from being able to regard microbiota modulation as a viable therapeutic strategy in the management of AA in our clinical practice. To date, there exists a notable gap between microbiome research and its application in clinical practice, not only concerning immune-mediated dermatological conditions such as AA but also across the broader medical field. Despite a growing interest among patients in microbiome-based diagnostic and therapeutic interventions, the majority of research findings have yet to be integrated into clinical practice. To address these challenges, the formation of multidisciplinary teams comprising microbiome clinicians, microbiologists, immunologists, nutritionists, and other specialists, alongside the utilization of existing diagnostic and therapeutic resources (such as microbiota sequencing tools, stool banks, and fecal microbiota transplantation centers) that are progressively enhancing in quality, could prove beneficial. Furthermore, the development of networking and training facilities will be essential. These changes would cause a paradigm shift in the clinical and scientific communities, encouraging the use of diagnostic and therapeutic microbiome technologies in clinical practice. In conclusion, while current therapeutic strategies for AA remain limited and often unsatisfactory, the gut microbiota represents a novel and promising avenue for future research. A deeper understanding of its role could not only elucidate key mechanisms in AA pathogenesis but also unlock innovative and potentially more effective therapeutic options.

## Figures and Tables

**Figure 1 biomedicines-13-01379-f001:**
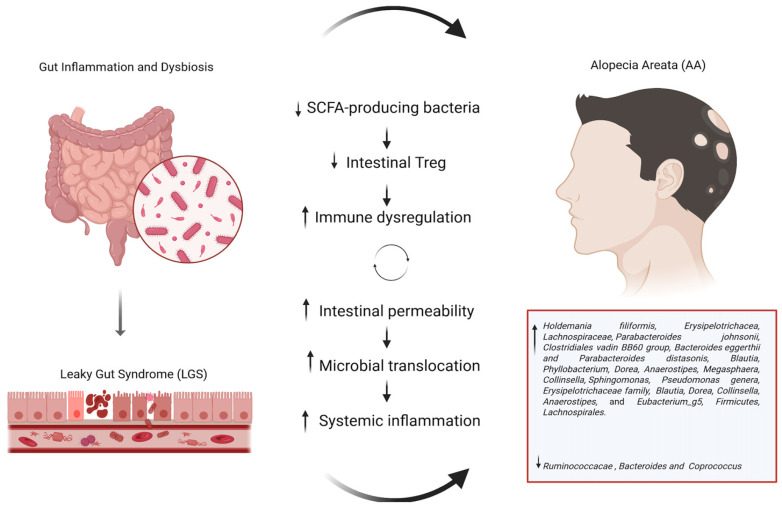
Overview of gut microbiome-mediated mechanism involved in the pathogenesis of AA.

**Table 1 biomedicines-13-01379-t001:** Overview of cross-sectional studies intended to identify characteristics in the gut microbial composition of patients with alopecia and healthy controls.

Study	Country	Study Population	Control Population	Analysis Methodology	Quantitative Differences	Qualitative Differences
Moreno-Arrones 2020 [66]	Spain	15 patients affected by alopecia universalis Mean age: 42.3 years. Disease duration > 6 months: 13 patients	15 healthy controls Mean age: 37.9 years	16SrRNA of stool samples	No significant differences in alpha and beta-diversity	Enriched presence of *Holdemania filiformis*, *Erysipelotrichacea*, *Lachnospiraceae*, *Parabacteroides johnsonii*, *Clostridiales vadin BB60 group*, *Bacteroides eggerthii*, and *Parabacteroides distasonis*
Lu 2021 [67]	China	33 patients affected by alopecia areata Mean age: 33.8 years	35 healthy controls Mean age: 37.3 years	16SrRNA of stool samples	No significant differences in alpha diversity	Enriched presence of *Blautia*, *Anaerostipes*, *Erysipelotrichaceae* (uncultured), *Dorea*, *Collinsella*, *Megasphaera*, and *Achromobacter*
Rangu 2021 [68]	USA	41 children with alopecia areata Age: 4–17 years	41 healthy siblings Age: 4–17 years	Shotgun metagenomic sequencing of stool samples	No significant differences in alpha diversity	Decrease in relative abundance of *Ruminococcus bicirculans*
Lee 2024 [69]	Korea	19 patients affected by alopecia areata Mean age: 44.6 years Mean disease duration: 12 months	20 healthy controls Mean age: 50.5 years	16SrRNA of stool samples	No significant differences in alpha diversity	Enriched presence of *Blautia*, *Dorea*, *Collinsella*, *Anaerostipes*, and *Eubacterium_g5*; decrease in relative abundance of *Ruminococcacae* and *Bacteroides*
Nikoloudaki 2024 [70]	Italy	24 patients affected by alopecia areata Mean age: 40 years	18 healthy controls Mean age: 45 years	16SrRNA of stool samples	No significant differences in alpha diversity	Enriched presence of *Firmicutes*, *Lachnospirales*, and *Blautia*. Descrease in relative abundance of *Coprococcus*

**Table 2 biomedicines-13-01379-t002:** Overview of Mendelian randomization analyses investigating the potential causal relationship between gut microbiota and alopecia areata.

Study	Country	Protective Role	Causative Role
Bi 2024 [71]	China	*Bacteroides A plebeius* *Brevibacillaceae* *Phascolarctobacterium sp003150755* *Provencibacterium massiliense* *UBA1066* *Provencibacterium*	*CAG-433,* *Chromatiales* *Clostridium E sporosphaeroides* *Comamonas*, *Cyanobacteria* *Dorea* *Lactobacillus B salivarius* *Leptospira* *Parachlamydiales* *UBA1777 sp900319835*
Xu 2024 [72]	China	*Butyricimonas* *Enterorhabdus* *Eubacterium* *Phascolarctobacterium*	*Ruminococcaceae UCG003*

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
