# Peer review of "The Multi-Faceted Role of Gut Microbiota in Alopecia Areata"

_biomedicines, 2025, doi:10.3390/biomedicines13061379_

Round 1
Reviewer 1 Report
Comments and Suggestions for Authors
The manuscript could be enhanced by the inclusion of the following:
- While the gut–skin axis is discussed, a more detailed breakdown of
specific immune signaling pathways and microbial metabolites implicated in AA would enrich the mechanistic depth of the review. - More critical discussion on the limitations and challenges of
translating microbiota-targeted therapies into clinical practice would be valuable—especially in light of the heterogeneity in current results. - While the article concludes with a call for further research, more specific
proposals for study designs (e.g., longitudinal cohorts, randomized controlled trials) and
standardized microbiome profiling methods would strengthen this section. Overall, this review is a well-written and relevant contribution to the field. With minor additions to improve structure and depth, it can serve as a foundational resource for researchers exploring the intersection of dermatology, immunology, and microbiome science.
Author Response
Rome, 01 June 2025
Dear Reviewer,
We would like to thank you for the careful assessment of our paper and for your precious comments, that improved the quality of our paper. We have done our best to address comments satisfactorily and hope that you will appreciate the revised version of the paper. Please find below a point-by-point response to your comments.
Best regards
Andrea Severino and Gianluca Ianiro, on behalf of all co-authors
POINT-BY-POINT RESPONSE
- While the gut–skin axis is discussed, a more detailed breakdown of specific immune signaling pathways and microbial metabolites implicated in AA would enrich the mechanistic depth of the review.
R: Dear Reviewer, I sincerely appreciate your insightful comments. I concur with your suggestion that a more comprehensive analysis of specific immune signaling pathways and microbial metabolites associated with AA would enhance the review. However, based on the current literature, the mechanisms that have been more precisely elucidated over time, linking the development of AA with the gut microbiota, are those previously mentioned:
- primarily, the disruption of immunotolerance mechanisms due to an altered Th17 to Treg lymphocyte ratio. This mechanism appears to be connected to the gut microbiota through the altered production of SCFAs, as discussed in section 4.3. (Lee, J.H.; Shin, J.H.; Kim, J.Y.; Ju, H.J.; Kim, G.M. Exploring the Role of Gut Microbiota in Patients with Alopecia Areata. Int. J. Mol. Sci. 2024, 25, 4256. https://doi.org/10.3390/ ijms25084256; Nikoloudaki, O.; Pinto, D.; Acin Albiac, M.; Celano, G.; Da Ros, A.; De Angelis, M.; Rinaldi, F.; Gobbetti, M.; Di Cagno, R. Exploring the Gut Microbiome and Metabolome in Individuals with Alopecia Areata Disease. Nutrients 2024, 16, 858. https://doi.org/10.3390/nu16060858)
- Secondly, there is a hypothesis that low-grade systemic inflammation, related to altered membrane permeability and the translocation of proinflammatory bacterial metabolites or components, such as LPS, into the circulation, may play a role. This mechanism is also addressed in section 4.3. (Lee, J.H.; Shin, J.H.; Kim, J.Y.; Ju, H.J.; Kim, G.M. Exploring the Role of Gut Microbiota in Patients with Alopecia Areata. Int. J. Mol. Sci. 2024, 25, 4256. https://doi.org/10.3390/ ijms25084256; Severino, A. et al. The microbiome-driven impact of western diet in the development of noncommunicable chronic dis-orders. Best Practice & Research Clinical Gastroenterology 72, 101923 (2024)
Additional mechanisms remain either unknown or insufficiently characterized, rendering the contribution of the gut microbiota to AA a field with existing gaps that we hope will be addressed in future research, building on the foundational evidence presented in this review. If you are aware of any articles or studies that could provide further insights in this area, we would be grateful and eager to incorporate them into the text to enhance its scientific quality.
- More critical discussion on the limitations and challenges of translating microbiota-targeted therapies into clinical practice would be valuable—especially in light of the heterogeneity in current results.
R: We have included a more critical discussion in section 7
- While the article concludes with a call for further research, more specific proposals for study designs (e.g., longitudinal cohorts, randomized controlled trials) and standardized microbiome profiling methods would strengthen this section.
R: We have implemented section 7 in order to address your kind suggestion
Reviewer 2 Report
Comments and Suggestions for Authors
Dear author,
Few points to clarify and improve article’s structure:
- There is repetition of ideas and concepts, particularly regarding SCFAs and their immunological effects, across different sections (4.1 and 4.3). Consolidating core concepts would help prevent dispersion and improve clarity.
-
5.1 and 5.2 mix the review of studies with the interpretation of data, which undermines the flow and coherence of the text. A clearer separation on clinical study results (by year and/or country) from the critical analysis of the findings may be necessary.
- Table 1. Please review data: ...Average age (Children/young adults/elderly), use of immunosuppressive medication, disease duration → these data are important for interpreting gut microbiota composition. if this is not avalable, just add a comment on text.
-
Standardize bacterial genus names (e.g., Blautia, Collinsella, etc.) and order ( most frequent bacteria could be cited first, so it is easier to compare the studies)
-
After Table 1, include an interpretive paragraph on limitations of the studies, potential biases (small sample sizes, geographical variation, sequencing methodology, etc.).
- Abrupt transition between item 4 and 5. Conduct the reader: “In light of these pathophysiological observations, clinical hypotheses and preliminary data... emerged exploring the correlation between gut dysbiosis and cases of alopecia areata.”
-
The conclusion paragraph should alert the uncertainties that still remain; importance of longitudinal and randomized human studies and risk of premature interpretation based on case reports or preclinical data.
Author Response
Rome, 01 June 2025
Dear Reviewer,
We would like to thank you for the careful assessment of our paper and for your precious comments, that improved the quality of our paper. We have done our best to address comments satisfactorily and hope that you will appreciate the revised version of the paper. Please find below a point-by-point response to your comments.
Best regards
Andrea Severino and Gianluca Ianiro, on behalf of all co-authors
POINT-BY-POINT RESPONSE
- There is repetition of ideas and concepts, particularly regarding SCFAs and their immunological effects, across different sections (4.1 and 4.3). Consolidating core concepts would help prevent dispersion and improve clarity.
R: The text has been modified (section 4.1 in particular) in order to avoid repetitions
- 1 and 5.2 mix the review of studies with the interpretation of data, which undermines the flow and coherence of the text. A clearer separation on clinical study results (by year and/or country) from the critical analysis of the findings may be necessary.
R: The text has been modified in order to achieve a clearer separation between study results and critical analysis of the findings. For every study the year has been reported in the text, while the countries are reported in table 1.
- Table 1. Please review data: ...Average age (Children/young adults/elderly), use of immunosuppressive medication, disease duration → these data are important for interpreting gut microbiota composition. if this is not avalable, just add a comment on text.
R: Table 1 has been modified adding the suggested informations, when available
- Standardize bacterial genus names (e.g., Blautia, Collinsella, etc.) and order ( most frequent bacteria could be cited first, so it is easier to compare the studies)
R: Table 1 has been modified and bacterial genus are now ordered on frequency basis.
- After Table 1, include an interpretive paragraph on limitations of the studies, potential biases (small sample sizes, geographical variation, sequencing methodology, etc.).
R: An interpretative paragraph on limitation of the studies has been added
- Abrupt transition between item 4 and 5. Conduct the reader: “In light of these pathophysiological observations, clinical hypotheses and preliminary data... emerged exploring the correlation between gut dysbiosis and cases of alopecia areata.”
R: Modifications have been made in paragraph 5 in order to avoid abrupt transition
- The conclusion paragraph should alert the uncertainties that still remain; importance of longitudinal and randomized human studies and risk of premature interpretation based on case reports or preclinical data.
R: Conclusion has been modified according to suggestions
Reviewer 3 Report
Comments and Suggestions for Authors
Minor Revisions Request:
The correlation hypothesis between the reduction of T lymphocytes and SFCA should be further explored.
A further study would also be useful on the relationship between alopecia areata and leaky gut, in particular of the adhesion proteins involved, not only of the zonulins of the TJ but also of the connexins present in the intestinal epithelium, forming gap-junctions.
We also ask you to present further information on the approach of treatment with Fecal Microbiota Transplantation.
The tables presented are dispersive; they could be reordered.
In the text, there are some typos.
Author Response
Rome, 01 June 2025
Dear Reviewer,
We would like to thank you for the careful assessment of our paper and for your precious comments, that improved the quality of our paper. We have done our best to address comments satisfactorily and hope that you will appreciate the revised version of the paper. Please find below a point-by-point response to your comments.
Best regards
Andrea Severino and Gianluca Ianiro, on behalf of all co-authors
POINT-BY-POINT RESPONSE
- The correlation hypothesis between the reduction of T lymphocytes and SFCA should be further explored.
R: More details about the correlation between reduction of T lymphocytes and SFCA have been added
- A further study would also be useful on the relationship between alopecia areata and leaky gut, in particular of the adhesion proteins involved, not only of the zonulins of the TJ but also of the connexins present in the intestinal epithelium, forming gap-junctions.
R: Thank you for your kind suggestion, another study investigating the role of claudin-3 in AA has been included in section 4.3
- We also ask you to present further information on the approach of treatment with Fecal Microbiota Transplantation.
R: Dear Reviewer, I concur with your observation that incorporating additional information on the correlation between Fecal Microbiota Transplantation (FMT) and alopecia areata would enhance the value of our study. However, to the best of our knowledge, the only literature on this topic comprises the previously mentioned case reports by the groups of Xie and Rebello. These studies involve a limited patient sample, rendering any further interpretation of the data precarious. Consequently, we have refrained from extensively discussing the modalities and mechanisms by which FMT might serve as a therapeutic alternative for this condition, in order to avoid engaging in speculative discourse unsupported by sufficient data. Should you be aware of any other studies on this subject that could enrich our work, we would be grateful to incorporate them into the manuscript.
- The tables presented are dispersive; they could be reordered.
R: Tables has been modified according to yours and other reviewers’ suggestions
- In the text, there are some typos.
R: We have looked for typos in the text and corrected the ones we found